# Safety and effectiveness of monovalent COVID-19 mRNA vaccination and risk factors for hospitalisation caused by the omicron variant in 0.8 million adolescents: A nationwide cohort study in Sweden

**Peter Nordström**[1,2]*, **Marcel Ballin**[2], **Anna Nordström**[3,4]

**1** Department of Public Health and Caring Sciences, Clinical Geriatrics, Uppsala University, Uppsala, Sweden, **2** Department of Community Medicine and Rehabilitation, Unit of Geriatric Medicine, Umeå University, Umeå, Sweden, **3** Department of Public Health and Clinical Medicine, Section of Sustainable Health, Umeå University, Umeå, Sweden, **4** School of Sport Sciences, UiT the Arctic University of Norway, Tromsø, Norway

* peter.nordstrom@pubcare.uu.se

**Data Availability Statement:** The data underlying the findings of the present study were used under ethical approval and are publicly unavailable according to regulations under Swedish law. Researchers who are interested in obtaining the data can contact Statistics Sweden via

## Abstract

### Background

Real-world evidence on the safety and effectiveness of Coronavirus Disease 2019 (COVID-19) vaccination against severe disease caused by the omicron variant among adolescents is sparse. In addition, evidence on risk factors for severe COVID-19 disease, and whether vaccination is similarly effective in such risk groups, is unclear. The aim of the present study was therefore to examine the safety and effectiveness of monovalent COVID-19 mRNA vaccination against COVID-19 hospitalisation, and risk factors for COVID-19 hospitalisation in adolescents.

### Methods and findings

A cohort study was conducted using Swedish nationwide registers. The safety analysis included all individuals in Sweden born between 2003 and 2009 (aged 11.3 to 19.2 years) given at least 1 dose of monovalent mRNA vaccine ($N = 645,355$), and never vaccinated controls ($N = 186,918$). The outcomes included all-cause hospitalisation and 30 selected diagnoses until 5 June 2022. The vaccine effectiveness (VE) against COVID-19 hospitalisation, and risk factors for hospitalisation, were evaluated in adolescents given 2 doses of monovalent mRNA vaccine ($N = 501,945$), as compared to never vaccinated controls ($N = 157,979$), for up to 5 months follow-up during an omicron predominant period (1 January 2022 to 5 June 2022). Analyses were adjusted for age, sex, baseline date, and whether the individual was born in Sweden. The safety analysis showed that vaccination was associated with 16% lower (95% confidence interval (CI) [12, 19], $p < 0.001$) risk of all-cause hospitalisation, and with marginal differences between the groups regarding the 30 selected diagnoses. In the VE analysis, there were 21 cases (0.004%) of COVID-19 hospitalisation among

mikrodata@scb.se (https://www.scb.se/en/services/ordering-data-and-statistics/ordering-microdata/), National Board of Health and Welfare via registerservice@socialstyrelsen.se (https://bestalladata.socialstyrelsen.se), and Public Health Agency of Sweden via info@folkhalsomyndigheten.se (https://www.folkhalsomyndigheten.se/the-public-health-agency-of-sweden/, and https://www.folkhalsomyndigheten.se/smittskydd-beredskap/utbrott/aktuellautbrott/covid-19/statistik-och-analyser/sars-cov-2-virusvarianter-av-sarskild-betydelse/).

**Funding:** The author(s) received no specific funding for this work.

**Competing interests:** The authors have declared that no competing interests exist.

**Abbreviations:** CI, confidence interval; COVID-19, Coronavirus Disease 2019; HR, hazard ratio; NNV, number needed to vaccinate; OR, odds ratio; SARS-CoV-2, Severe Acute Respiratory Syndrome Coronavirus 2; VE, vaccine effectiveness.

2-dose recipients and 26 cases (0.016%) among controls, resulting in a VE of 76% (95% CI [57, 87], $p < 0.001$). Predominant risk factors for COVID-19 hospitalisation included previous infections (bacterial infection, tonsillitis, and pneumonia) (odds ratio [OR]: 14.3, 95% CI [7.7, 26.6], $p < 0.001$), and cerebral palsy/development disorders (OR: 12.7, 95% CI [6.8, 23.8], $p < 0.001$), with similar estimates of VE in these subgroups as in the total cohort. The number needed to vaccinate with 2 doses to prevent 1 case of COVID-19 hospitalisation was 8,147 in the total cohort and 1,007 in those with previous infections or developmental disorders. None of the individuals hospitalised due to COVID-19 died within 30 days. Limitations of this study include the observational design and the possibility of unmeasured confounding.

## Conclusions

In this nationwide study of Swedish adolescents, monovalent COVID-19 mRNA vaccination was not associated with an increased risk of any serious adverse events resulting in hospitalisation. Vaccination with 2 doses was associated with a lower risk of COVID-19 hospitalisation during an omicron predominant period, also among those with certain predisposing conditions who should be prioritised for vaccination. However, COVID-19 hospitalisation in the general population of adolescents was extremely rare, and additional doses in this population may not be warranted at this stage.

## Author summary

### Why was this study done?

**Evidence before this study**

➢ There is limited evidence on the effectiveness of Coronavirus Disease 2019 (COVID-19) vaccination against severe outcomes during the omicron era among adolescents.

➢ In addition, there is lack of data on whether certain groups of adolescents are at greater risk of severe COVID-19 and should be prioritised in vaccination programs, and whether vaccination is equally effective in such risk groups.

➢ Regarding safety, some studies have indicated a link between COVID-19 mRNA vaccination and increased risk of myocarditis and pericarditis in young men.

### What did the researchers do and find?

➢ Using Swedish nationwide health registers, a nationwide cohort of adolescents were followed for up to 5 months during the omicron era to evaluate the safety and effectiveness of monovalent COVID-19 mRNA vaccination against COVID-19 hospitalisation, and risk factors for COVID-19 hospitalisation.

➢ Adolescents vaccinated with at least 1 dose of a monovalent COVID-19 mRNA vaccine did not have a higher risk of hospitalisation for any diagnosis, as compared to unvaccinated adolescents.

➤ In contrast, individuals vaccinated with 2 doses of vaccine had 76% lower risk of being hospitalised due to COVID-19, although only about 7 individuals in 100,000 were hospitalised due to COVID-19 during follow-up.

➤ There were specific risk factors for COVID-19 hospitalisation, including previous infections and different development disorders, which increased the risk of COVID-19 hospitalisation more than 10-fold. Vaccine effectiveness among these individuals was similar as in the rest of the cohort.

## What do these findings mean?

➤ Although monovalent COVID-19 mRNA vaccination appears safe and associated with reduced risk of COVID-19 hospitalisation, the risk of severe COVID-19 seems to be extremely low in the general population of adolescents.

➤ Administration of additional vaccine doses to the general population of adolescents may not be warranted at this stage of the pandemic.

➤ In contrast, individuals with a high risk for severe COVID-19 should be prioritised for vaccination.

## Introduction

The omicron variant of Severe Acute Respiratory Syndrome Coronavirus 2 (SARS-CoV-2) led to a surge in cases among young people during 2022 when several countries lifted or alleviated their restrictions [1]. Although clinical trials showed that the BNT162b2 and mRNA-1273 vaccines had acceptable safety profiles in adolescents and reduced the risk of infection in the short term [2,3], these trials were conducted before the emergence of omicron. Limited sample sizes also hindered the evaluation of the level of protection against severe Coronavirus Disease 2019 (COVID-19), such as hospitalisation, which could be estimated using large-scale observational studies. These studies also offer the possibility of investigating any potential links between vaccination and rare serious adverse events, such as myocarditis [4,5].

Currently, there is limited data on vaccine effectiveness (VE) against severe COVID-19 caused by the omicron variant among adolescents. Two case–control studies found that 2 doses of the BNT162b2 mRNA vaccine had about 80% VE against COVID-19 hospitalisation or death in adolescents during the omicron era [6,7]. However, given their study design, it is difficult to determine how common severe disease is during the omicron era. In addition, little is known about whether certain groups of adolescents should be prioritised for vaccination because of a higher risk of severe COVID-19, and whether vaccination has a similarly protective effect in such risk groups. Moreover, although a third dose, also known as a booster dose, may increase protection against symptomatic omicron infection in adolescents [8,9], the risk of severe COVID-19 after a third dose relative to after the second dose is unclear.

In the present study, we used Swedish nationwide registers to evaluate, among adolescents, (1) the risk of hospitalisation from any cause following monovalent COVID-19 mRNA vaccination, and (2) the effectiveness of monovalent COVID-19 mRNA vaccination against hospitalisation due to COVID-19 and risk factors for COVID-19 hospitalisation during an omicron predominant period.

## Methods

### Study design and cohorts

This nationwide, retrospective cohort study was approved by the Swedish Ethical Review Authority (number 00094/2021). There was no prospective written analysis plan for the present study, and the construction of the models and analyses were data driven. This study is reported as per the Strengthening the Reporting of Observational Studies in Epidemiology (STROBE) guideline (S1 STROBE Checklist).

The cohort considered for inclusion was compiled by Statistics Sweden (www.SCB.se); the government agency in charge of nationwide statics in different areas and covering the total population of Sweden. The Public Health Agency of Sweden provided data for all individuals born 2003 to 2009 who were given at least 1 dose of monovalent COVID-19 mRNA vaccine or had a documented SARS-CoV-2 infection until March 2022 ($N$ = 692,419). To vaccinated individuals, Statistics Sweden matched 1 control individual on birth year, sex, and municipality. Controls could not have received a first dose of vaccine at the date when the corresponding vaccinated individual had received 2 doses of vaccine. The cohort was updated with vaccination data and SARS-CoV-2 infections until 2 June 2022. Because 1 control could be matched to several vaccinated individuals, the total eligible cohort consisted of 832,273 individuals, of whom 645,355 had been vaccinated with at least 1 dose, 600,721 had been vaccinated with 2 doses, and 60,391 had been vaccinated with at least 3 doses (Fig 1). In a first set of analyses,

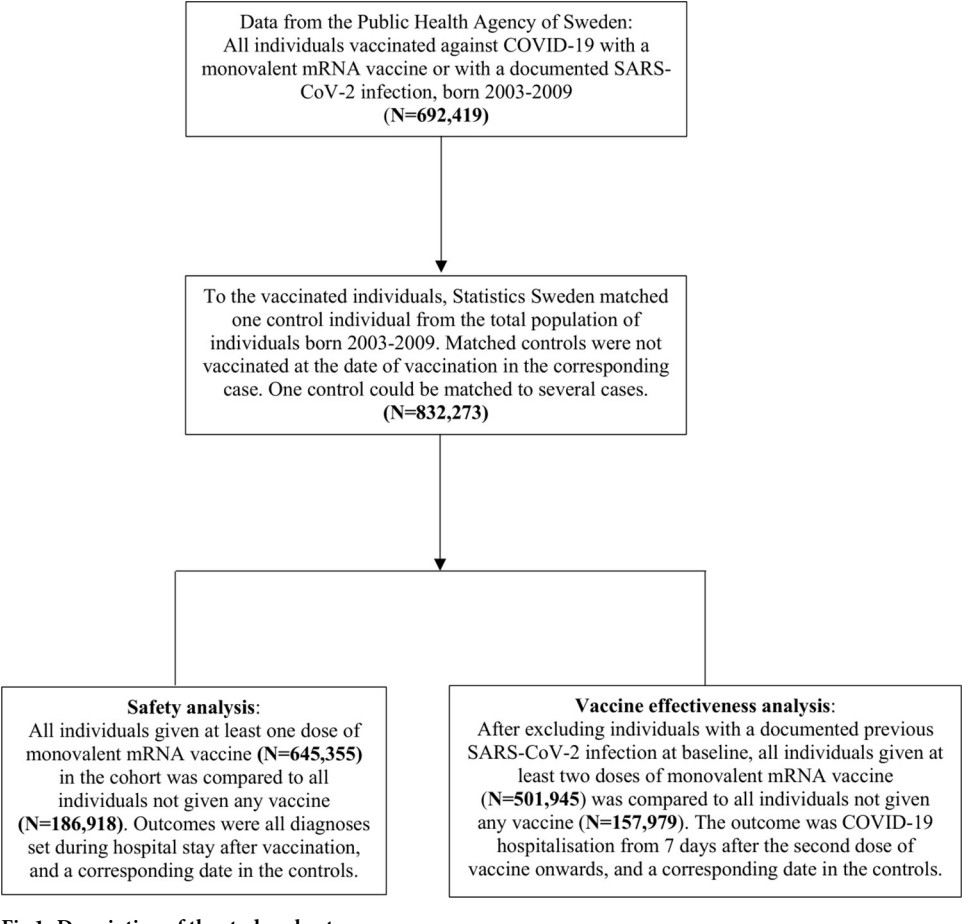

**Fig 1. Description of the study cohort.**

named the safety analysis, all diagnoses set during hospitalisation were evaluated in all individuals given at least 1 dose of vaccine during follow-up (N = 645,355) as compared to individuals never vaccinated during follow-up (N = 186,918). In a second set of analyses, VE against COVID-19 hospitalisation was evaluated by comparing individuals given at least 2 doses of vaccine (N = 501,945) to individuals never vaccinated during follow-up (N = 157,979), excluding all individuals with a previous documented SARS-CoV-2 infection (Fig 1). Data on individuals vaccinated against COVID-19 and data on documented SARS-CoV-2 infections were collected from the Swedish Vaccination Register and the SmiNet register, respectively. Both these registers are managed by the Public Health Agency of Sweden and all healthcare providers in Sweden were obliged to report to these registers according to law [10,11].

## Outcomes

In the safety analysis, based on all diagnoses set in the cohort during inpatient hospital stay until 5 June 2022, results are presented for all-cause hospitalisation and for 30 different diagnoses. The 30 specific diagnoses were selected based on their incidence and general interest given previous reports of links between certain diagnoses and COVID-19 mRNA vaccination [4,12,13]. Only the first diagnosis was evaluated for each individual, and individuals with this diagnosis at baseline were therefore excluded from the prospective analyses.

In the VE analysis, the primary outcome was a main diagnosis of COVID-19 (thus, "due to" COVID-19 rather than "with" COVID-19), set during inpatient hospital stay from 1 January 2022 until 5 June 2022. The secondary outcome was documented SARS-CoV-2 infection of any severity from the SmiNet register, from 1 January 2022 until 28 February 2022, in line with the changes made to testing guidelines in Sweden. For these 2 outcomes, only cases that occurred 7 days after the second dose of vaccine and onwards were counted to ensure the full effect of vaccination. The start of 1 January was selected on the basis that the omicron variant was first documented in Sweden on 29 November 2021 [14], and by early January 2022, it represented >90% of sequenced cases (S1 Table).

Data on all diagnoses used in this study were obtained from the National Patient Register [15] using 10th revision International Classification of Disease codes, starting from 1 January 2016 earliest until 5 June 2022 latest. The definition and diagnostic codes for all diagnoses considered in the safety analysis and for the analysis of risk factors for COVID-19 hospitalisation are shown in Table 1. The positive predictive value for diagnoses set within the National Patient Register differs but has generally been found to be between 85% and 95%, although sensitivity is often lower [15]. Finally, data on death due to COVID-19 or influenza, defined as death within 30 days after the diagnosis, were obtained from the National Cause of Death Register [16].

## Statistical analysis

In the safety analysis, diagnoses were evaluated before and after the first dose of vaccine in 645,355 individuals given at least 1 dose of vaccine and in 186,918 controls who were unvaccinated during the follow-up time. The baseline date in vaccinated individuals was the date of vaccination with the first dose. In the controls, a baseline date was randomly assigned based on the mean baseline date and standard deviation among the vaccinated individuals (10 October 2021 ± 54 days). Student t tests and chi-squared tests were used to compare the prevalence of different variables at baseline. To estimate hazard ratios (HRs) for all-cause hospitalisation and for the 30 different diagnoses during follow-up, Cox regression models were used. Individuals were censored on the date of the diagnosis of interest, death, or end of follow-up (5 June 2022), whichever came first.

**Table 1. Baseline characteristics of the individuals at date of the first dose of vaccine and in never vaccinated individuals.**

|  | Vaccinated with ≥1 dose (N = 645,355) | Never vaccinated (N = 186,918) | p |
|---|---|---|---|
| Age, years (standard deviation) | 15.4 (1.9) | 14.8 (2.0) | < 0.001 |
| Female sex | 318,983 (49.4) | 85,798 (45.9) | < 0.001 |
| Born in Sweden | 567,298 (87.9) | 139,213 (74.5) | < 0.001 |
| Mean baseline date | 10 October 2021 | 10 October 2021 | < 0.001 |
| Previous SARS-CoV-2 infection | 91,934 (14.2) | 20,896 (11.2) | < 0.001 |
| **Previous diagnoses (ICD-10 code)** |  |  |  |
| Any diagnosis | 25,686 (4.0) | 7,515 (4.0) | < 0.001 |
| Gastroenteritis (A09) | 125 (0.019) | 37 (0.020) | 0.32 |
| Sepsis (A41) | 45 (0.007) | 17 (0.009) | 0.54 |
| Erysipelas (A46) | 25 (0.004) | 9 (0.005) | 0.93 |
| Bacterial infection unspecified (A49) | 21 (0.003) | 8 (0.004) | 0.29 |
| Mononucleosis (B27) | 224 (0.034) | 38 (0.020) | < 0.001 |
| Virus infection, unspecified (B34) | 114 (0.018) | 34 (0.018) | 0.59 |
| Chronic lymphocytic leukemia (C91) | 30 (0.005) | 13 (0.007) | 0.07 |
| Iron deficiency anemia (D50) | 93 (0.014) | 40 (0.021) | 0.16 |
| Thrombocytopenia (D69) | 44 (0.007) | 14 (0.007) | 0.19 |
| Agranulocytosis (D70) | 24 (0.004) | 8 (0.004) | 0.93 |
| Alcohol dependency (F10) | 975 (0.15) | 313 (0.17) | 0.48 |
| Depressive episode (F32) | 881 (0.14) | 211 (0.11) | < 0.001 |
| Anxiety state, unspecified (F41) | 586 (0.09) | 185 (0.10) | 0.38 |
| Allergy or anaphylactic shock (T78) | 177 (0.027) | 58 (0.031) | 0.35 |
| Anorexia nervosa (F50) | 571 (0.088) | 107 (0.092) | < 0.001 |
| Epilepsy (G40) | 318 (0.049) | 113 (0.060) | 0.004 |
| Otitis media (H66) | 22 (0.003) | 7 (0.004) | 0.11 |
| Myocarditis (I40) | 50 (0.008) | 19 (0.010) | 0.75 |
| Pericarditis (I30) | 26 (0.004) | 10 (0.005) | 0.62 |
| Sinusitis (J01) | 39 (0.006) | 17 (0.009) | 0.13 |
| Tonsillitis (J03) | 166 (0.026) | 38 (0.020) | 0.67 |
| Chronic tonsillitis (J35) | 208 (0.032) | 58 (0.031) | 0.58 |
| Upper respiratory infection (J06) | 64 (0.010) | 18 (0.010) | 0.87 |
| Pneumonia (J15 and J18) | 151 (0.023) | 40 (0.021) | 0.65 |
| Peritonsillar abscess (J36) | 118 (0.018) | 37 (0.020) | 0.82 |
| Appendicitis (K35) | 2,071 (0.32) | 544 (0.29) | < 0.001 |
| Crohn's disease (K50) | 107 (0.017) | 39 (0.021) | 0.15 |
| Cutaneous abscess (L02) | 72 (0.011) | 29 (0.016) | 0.09 |
| Nephritis (N10) | 252 (0.039) | 53 (0.028) | 0.10 |
| Traumatic brain injury (S06) | 1,212 (0.19) | 293 (0.16) | < 0.001 |
| **Combined diagnoses** |  |  |  |
| Cerebral palsy/development disorders (G80, F73, F82, F84, Q02) | 23,093 (3.6) | 8,307 (4.4) | < 0.001 |
| Selected infections (A49, J03, J15, J18, J35) | 21,208 (3.3) | 7,158 (3.8) | < 0.001 |

All data are shown as number (percentage) unless stated otherwise.

All previous diagnoses were main diagnoses set during inpatient hospital stay from 1 January 2020 earliest until 5 June 2022 latest as obtained from the National Patient Register, with the exception of the diagnoses listed under "combined diagnoses," for which diagnoses set from 1 January 2016 and later was used and including secondary outpatient care.

In the VE analysis, the proportional hazards assumption was not met; hence, logistic regression was used to estimate odds ratios (ORs) for the primary outcome of COVID-19 hospitalisation (from 1 January 2022 until 5 June 2022), and for the secondary outcome of a SARS-CoV-2 infection (from 1 January 2022 until 28 February 2022). The ORs obtained were used to estimate VE as 1 minus the OR × 100. In all analyses, the first model was unadjusted and the second model was adjusted for baseline date, age, sex, and whether the individual was born in Sweden or not. Data underlying these covariates were retrieved from Statistics Sweden [17]. To investigate whether VE differed by the covariates, interaction analyses were performed using product terms created by multiplying the variable coding for vaccination status at baseline (vaccinated/unvaccinated) by each respective covariate, which was added to the logistic regression model. Given that the interaction term was statistically significant $(p < 0.05)$ for the baseline date, VE was also estimated in subgroups according to this covariate. The number needed to vaccinate (NNV) with 2 doses to prevent 1 case of COVID-19 hospitalisation during follow-up was estimated as the inverse of the absolute risk difference between the groups (vaccinated/unvaccinated).

Finally, a sensitivity analysis using a negative control outcome was conducted to explore the potential risk of bias due to unmeasured confounding [18]. Here, a logistic regression model was performed, comparing individuals given at least 2 doses of COVID-19 mRNA vaccine compared to never vaccinated individuals concerning the outcome of hospitalisation due to influenza from 1 January 2022 until 5 June 2022. All analyses were performed in SPSS v29.0 for Mac (IBM, Armonk, New York, USA), and Stata v16.1 for Mac (Statcorp, College Station, Texas, USA). A two-sided $p$-value $< 0.05$ or ORs/HRs with 95% confidence intervals (CIs) not crossing one were considered statistically significant.

## Results

The total cohort comprised 832,273 adolescents born 2003 to 2009 (age 11.3 to 19.2 years), of whom 645,355 received at least 1 dose of COVID-19 mRNA vaccine and 186,918 never vaccinated individuals (controls). Almost 90% of the vaccinated individuals received BNT162b2 as a first dose, while the remaining received mRNA-1273. Baseline characteristics are shown in Table 1. Individuals that were never vaccinated were slightly younger, more often of male sex and born outside of Sweden, and less often diagnosed with a previous SARS-CoV-2 infection before baseline $(p < 0.001$ for all). Concerning other diagnoses at baseline, differences between the groups were marginal (Table 1).

### Serious adverse events after a first dose of vaccine

In the safety analysis ($N = 832,273$), there were a total of 19,580 all-cause hospitalisations among 14,266 individuals during follow-up. In vaccinated individuals, 1.69% ($N = 10,906$) were hospitalised at least once, compared to 1.80% ($N = 3,360$) in those never vaccinated (HR; 0.84, 95% CI [0.81, 0.88], $p < 0.001$; Table 2). There were marginal differences between the 2 groups in the 30 selected diagnoses (Table 2), although statistically significant associations in favour of vaccination were observed with respect to the risk of sepsis, thrombocytopenia, alcohol dependency, peritonsillar abscess, and Crohn's disease $(p < 0.05$ for all). None of the other associations were statistically significant after adjustment.

### Vaccine effectiveness against COVID-19 hospitalisation

In the analysis of VE against COVID-19 hospitalisation, 501,945 individuals vaccinated with 2 doses and 157,979 never vaccinated controls were included. Between 1 January 2022 and 5 June 2022, a total of 47 individuals (7 per 100,000) were hospitalised due to COVID-19. Of

**Table 2. Risk of hospitalisation for any cause and for 30 selected diagnoses in vaccinated individuals vaccinated compared to never vaccinated individuals.**

| | Vaccinated with ≥1 one dose (N = 645,355) | Never vaccinated (N = 186,918) | Unadjusted analyses | | Adjusted analyses | |
|---|---|---|---|---|---|---|
| Outcome | Number of cases (IR) | Number of cases (IR) | HR [95% CI] | p | HR [95% CI] | p |
| All-cause hospitalisation | 10,906 (71.9) | 3,360 (76.5) | 0.94 (0.91, 0.98) | 0.003 | 0.84 (0.81, 0.88) | < 0.001 |
| **Hospitalisation for selected diagnosis (ICD-10 code)** | | | | | | |
| Gastroenteritis (A09) | 58 (0.38) | 20 (0.45) | 0.85 (0.51, 1.42) | 0.54 | 0.99 (0.56, 1.66) | 0.89 |
| Sepsis (A41) | 5 (0.03) | 7 (0.16) | 0.20 (0.06, 0.64) | 0.006 | 0.17 (0.05, 0.56) | 0.003 |
| Erysipelas (A46) | 8 (0.05) | 2 (0.05) | 1.28 (0.27, 6.15) | 0.76 | 0.99 (0.18, 5.41) | 0.99 |
| Bacterial infection unspecified (A49) | 11 (0.07) | 3 (0.07) | 1.05 (0.29, 3.76) | 0.94 | 0.83 (0.23, 3.05) | 0.78 |
| Mononucleosis (B27) | 115 (0.76) | 26 (0.59) | 1.27 (0.83, 1.95) | 0.27 | 0.78 (0.50, 1.21) | 0.26 |
| Virus infection, unspecified (B34) | 42 (0.28) | 7 (0.16) | 1.74 (0.78, 3.86) | 0.18 | 1.52 (0.67, 3.43) | 0.32 |
| Chronic lymphocytic leukemia (C91) | 4 (0.03) | 4 (0.09) | 0.29 (0.07, 1.16) | 0.08 | 0.36 (0.08, 1.50) | 0.16 |
| Iron deficiency anemia (D50) | 38 (0.25) | 14 (0.32) | 0.79 (0.43, 1.46) | 0.45 | 0.88 (0.47, 1.66) | 0.70 |
| Thrombocytopenia (D69) | 8 (0.05) | 10 (0.23) | 0.18 (0.10, 0.65) | 0.004 | 0.21 (0.08, 0.58) | 0.002 |
| Agranulocytosis (D70) | 1 (0.007) | 1 (0.02) | - | | - | - |
| Alcohol dependency (F10) | 409 (2.69) | 123 (2.80) | 0.97 (0.79, 1.18) | 0.74 | 0.75 (0.61, 0.93) | 0.008 |
| Depressive episode (F32) | 363 (2.39) | 102 (2.32) | 1.03 (0.83, 1.29) | 0.77 | 0.88 (0.70, 1.11) | 0.28 |
| Anxiety state, unspecified (F41) | 341 (2.25) | 75 (1.70) | 1.40 (1.09, 1.80) | 0.009 | 0.92 (0.71, 1.19) | 0.51 |
| Allergy or anaphylactic shock (T78) | 64 (0.42) | 23 (0.52) | 0.81 (0.50, 1.30) | 0.38 | 0.70 (0.43, 1.15) | 0.16 |
| Anorexia nervosa (F50) | 197 (1.30) | 47 (1.07) | 1.21 (0.88, 1.67) | 0.23 | 1.08 (0.78, 1.50) | 0.64 |
| Epilepsy (G40) | 82 (0.54) | 33 (0.75) | 0.72 (0.48, 1.08) | 0.12 | 0.89 (0.57, 1.41) | 0.63 |
| Otitis media (H66) | 7 (0.05) | 3 (0.07) | 0.67 (0.17, 2.60) | 0.57 | 0.70 (0.16, 2.97) | 0.63 |
| Myocarditis (I40) | 86 (0.57) | 19 (0.43) | 1.31 (0.80, 2.15) | 0.29 | 0.99 (0.60, 165) | 0.98 |
| Pericarditis (I30) | 21 (0.14) | 3 (0.07) | 2.00 (0.60, 6.73) | 0.26 | 0.88 (0.26, 3.02) | 0.84 |
| Sinusitis (J01) | 15 (0.10) | 10 (0.23) | 0.43 (0.19, 0.96) | 0.04 | 0.64 (0.26, 1.58) | 0.33 |
| Tonsillitis (J03) | 90 (0.59) | 11 (0.25) | 2.34 (1.25, 4.38) | 0.008 | 1.51 (0.80, 2.86) | 0.21 |
| Chronic tonsillitis (J35) | 94 (0.59) | 18 (0.41) | 1.53 (0.92, 2.53) | 0.10 | 1.15 (0.68, 1.93) | 0.60 |
| Upper respiratory infection (J06) | 24 (0.16) | 4 (0.09) | 1.76 (0.61, 5.10) | 0.30 | 1.72 (0.56, 5.27) | 0.35 |
| Pneumonia (J15 and J18) | 68 (0.45) | 16 (0.36) | 1.26 (0.73, 2.17) | 0.41 | 1.47 (0.82, 2.64) | 0.20 |

(*Continued*)

**Table 2.** (Continued)

| Outcome | Vaccinated with ≥1 one dose (N = 645,355) | Never vaccinated (N = 186,918) | Unadjusted analyses | | Adjusted analyses | |
|---|---|---|---|---|---|---|
| | Number of cases (IR) | Number of cases (IR) | HR [95% CI] | p | HR [95% CI] | p |
| Peritonsillar abscess (J36) | 52 (0.34) | 17 (0.39) | 0.90 (0.52, 1.97) | 0.72 | 0.56 (0.32, 0.98) | 0.04 |
| Appendicitis (K35) | 665 (4.38) | 174 (3.96) | 1.11 (0.94, 1.32) | 0.21 | 1.04 (0.87, 1.23) | 0.69 |
| Crohn's disease (K50) | 24 (0.16) | 14 (0.32) | 0.49 (0.26, 0.96) | 0.04 | 0.46 (0.23, 0.92) | 0.03 |
| Cutaneous abscess (L02) | 10 (0.07) | 5 (0.11) | 0.58 (0.20, 1.69) | 0.32 | 0.59 (0.20, 1.69) | 0.35 |
| Nephritis (N10) | 105 (0.69) | 18 (0.41) | 1.69 (1.02, 2.79) | 0.04 | 1.22 (0.73, 2.03) | 0.45 |
| Traumatic brain injury (S06) | 307 (2.02) | 83 (1.89) | 1.08 (0.84, 1.37) | 0.56 | 1.09 (0.85, 1.41) | 0.50 |

CI, confidence interval; HR, hazard ratio. IR = incidence rates per 1 million person-days of follow-up. Adjusted analyses were adjusted for age, sex, baseline date, and whether the individual was born in Sweden.

these, 21 cases were among those vaccinated with 2 doses (0.004%), and 26 among unvaccinated (0.016%), resulting in a VE of 76% (95% CI [57, 87], $p < 0.001$) (Table 3). The NNV with 2 doses to prevent 1 case of COVID-19 hospitalisation was 8,147. For those with a second dose of vaccine earlier than 15 November, the VE was 69% (95% CI [29, 87], $p = 0.006$), compared to 87% (95% CI [66, 95], $p < 0.001$) for those with a second dose of vaccine 15 November 2021 and later ($p = 0.03$ for interaction). When comparing individuals vaccinated with 3 doses ($N = 41,225$) compared to those vaccinated with 2 doses ($N = 413,544$), only 12 individuals in total (2.6 per 100,000 individuals) were hospitalised due to COVID-19 during follow-up (VE; 13%, 95% CI [−354, 84], $p = 0.86$). There were no deaths within 30 days of hospitalisation among the 261 individuals hospitalised due to COVID-19 in the total cohort (832,273) since the beginning of the pandemic in January 2020.

**Table 3. VE against COVID-19 hospitalisation from 7 days onwards after a second dose of vaccine as compared to never vaccinated individuals from 1 January 2022 until 5 June 2022, and by time since vaccination and subgroups.**

| | Vaccinated with 2 doses | Never vaccinated | Unadjusted analyses | | Adjusted analyses | |
|---|---|---|---|---|---|---|
| | Number of cases (%) | Number of cases (%) | VE [95% CI] | p | VE [95% CI] | p |
| Total cohort (N = 659,924)[a] | 21 (0.004%) | 26 (0.016%) | 75 (55, 86) | < 0.001 | 76 (57, 87) | < 0.001 |
| Subgroups | | | | | | |
| Baseline date < 15 Nov 2021 (N = 287,871)[b] | 15 (0.007%) | 14 (0.018%) | 61 (19, 81) | 0.01 | 69 (29, 87) | 0.006 |
| Baseline date > 14 Nov 2021 (N = 372,053)[c] | 6 (0.002%) | 12 (0.015%) | 86 (63, 95) | < 0.001 | 87 (66, 95) | < 0.001 |
| Previously diagnosed with selected infections (N = 21,981)[d] | 4 (0.025%) | 11 (0.187%) | 87 (58, 96) | < 0.001 | 88 (58, 96) | < 0.001 |
| Previously diagnosed with development disorders (N = 25,832)[e] | 6 (0.032%) | 9 (0.124%) | 74 (27, 91) | 0.01 | 72 (20, 91) | 0.02 |

CI, confidence interval; VE, vaccine effectiveness.

Adjusted models were adjusted for age, baseline date, sex, and whether the individual was born in Sweden.

[a]Of which 501,945 were vaccinated and 157,979 never vaccinated.

[b]Of which 210,747 were vaccinated and 77,124 never vaccinated.

[c]Of which 291,198 were vaccinated and 87,187 never vaccinated.

[d]Of which 16,083 were vaccinated and 5,898 never vaccinated.

[e]Of which 18,553 were vaccinated and 7,279 never vaccinated.

### Risk factors for COVID-19 hospitalisation and vaccine effectiveness by subgroups

Of the 47 individuals hospitalised due to COVID-19 in the VE analysis, 28 (60%) had previously been hospitalised for another condition, compared to 66,483 (10%) of the individuals in the rest of the cohort. In addition, of those hospitalised, 15 (32%) had previously been diagnosed with an infection (bacterial infection, tonsillitis, and pneumonia), compared to 21,966 (3.3%) in the rest of the cohort, equal to an adjusted OR for COVID-19 hospitalisation of 14.3 (95% CI [7.7, 26.6], $p < 0.001$). The VE in this risk group was similar (VE; 88%, 95% CI [58, 96], $p < 0.001$), as in the total cohort, but with a slightly lower vaccination uptake (73.2% versus 76.2%). In addition, 15 (32%) of the individuals hospitalised due to COVID-19 had previously been diagnosed with cerebral palsy and/or different development disorders, compared to 25,817 individuals (3.9%) in the rest of the cohort, resulting in an adjusted OR for COVID-19 hospitalisation of 12.7 (95% CI [6.8, 23.8], $p < 0.001$). Again, VE in this subgroup was similar as in the total cohort (VE; 72%, 95% CI [20, 91], $p = 0.02$), but with a slightly lower vaccination coverage (71.7% versus 76.4%). The NNV with 2 doses to prevent 1 case of COVID-19 hospitalisation in the subgroup of individuals previously diagnosed with infections or development disorders ($N = 46,521$) was 1,007.

### Vaccine effectiveness against SARS-CoV-2 infection

The VE against SARS-CoV-2 infection of any severity was estimated among 488,441 individuals vaccinated with 2 doses compared to 153,882 never vaccinated controls. Between 1 January 2022 and 28 February 2022, there were 72,627 cases of confirmed SARS-CoV-2 infections. The VE varied by time since the last dose ($p < 0.001$; Table 4), with marginal VE in the total cohort (VE; 4%, 95% CI [3,4], $p < 0.001$), a low VE for individuals with a second dose no earlier than 1 November 2021 (VE; 27%, 95% CI [25, 29], $p < 0.001$), and a moderate VE for individuals with a second dose no earlier than 1 January 2022 (VE; 51%, 95% CI, [48, 55], $p < 0.001$).

**Table 4. VE against SARS-CoV-2 infection of any severity from 7 days onwards after a second dose of vaccine as compared to never vaccinated individuals from 1 January *2022* until 28 February 2022, and by baseline date.**

| | Vaccinated with 2 doses | Never vaccinated | Unadjusted analyses | | Adjusted analyses | |
|---|---|---|---|---|---|---|
| | Number of cases (%) | Number of cases (%) | VE [95% CI] | *p* | VE [95% CI] | *p* |
| Total cohort ($N = 642,323$)[a] | 55,101 (11.3%) | 17,530 (11.4%) | 1 (−1, 3) | 0.23 | 4 (3, 4) | < 0.001 |
| Subgroups by baseline date | | | | | | |
| ≥1 September 2021 ($N = 611,401$)[b] | 51,497 (11.0%) | 16,037 (11.3%) | 3 (2, 5) | < 0.001 | 12 (10, 13) | < 0.001 |
| ≥1 October 2021 ($N = 477,924$)[c] | 32,613 (9.2%) | 13,655 (11.0%) | 18 (17, 20) | < 0.001 | 17 (15, 19) | < 0.001 |
| ≥1 November 2021 ($N = 402,682$)[d] | 27,213 (8.8%) | 9,743 (10.5%) | 18 (16, 20) | < 0.001 | 27 (25, 29) | < 0.001 |
| ≥1 December 2021 ($N = 241,350$)[e] | 12,547 (6.8%) | 5,288 (9.1%) | 27 (24, 29) | < 0.001 | 41 (39, 43) | < 0.001 |
| ≥1 January 2022 ($N = 105,630$)[f] | 2,087 (2.6%) | 1,279 (4.8%) | 47 (43, 50) | < 0.001 | 51 (48, 55) | < 0.001 |

CI, confidence interval; VE, vaccine effectiveness.

Adjusted models were adjusted for age, baseline date, sex, and whether the individual was born in Sweden.

[a]Of which 488,441 were vaccinated and 153,882 never vaccinated.

[b]Of which 469,627 were vaccinated and 141,774 never vaccinated.

[c]Of which 354,304 were vaccinated and 123,620 never vaccinated.

[d]Of which 309,804 were vaccinated and 92,878 never vaccinated.

[e]Of which 183,462 were vaccinated and 57,888 never vaccinated.

[f]Of which 79,161 were vaccinated and 26,469 never vaccinated.

### Sensitivity analysis

The risk of hospitalisation due to influenza was evaluated in 600,721 individuals given at least 2 doses of COVID-19 vaccine compared to in 186,894 never vaccinated individuals. Between 1 January 2022 and 5 June 2022, a total of 47 individuals (6 per 100,000) were hospitalised due to influenza. The results showed that individuals vaccinated with 2 doses of vaccine did not experience a lower risk of hospitalisation due to influenza as compared to never vaccinated individuals (VE; −10%, 95% CI [−133, 48], $p = 0.81$), thus indicating an absence of important unmeasured confounding. Of the 117 individuals in the total cohort hospitalised for influenza since the start of the COVID-19 pandemic in January 2020, 1 individual died within 30 days of hospitalisation.

## Discussion

In this nationwide study of more than 0.8 million Swedish adolescents, vaccination with at least 1 dose of monovalent COVID-19 mRNA vaccine was not associated an increased risk of hospitalisation for any cause, and vaccination with 2 doses was associated with lower risk of COVID-19 hospitalisation during an omicron-predominant period. However, the absolute risk of COVID-hospitalisation was extremely low, except in subgroups of adolescents that had previously been diagnosed with infections, cerebral palsy, or other development disorders.

Evidence on the safety and effectiveness of COVID-19 mRNA vaccination in adolescents during the omicron-predominant period is limited. In this study, vaccination was not associated with an increased risk of hospitalisation from any cause, or for any of the 30 selected diagnoses. This suggests that COVID-19 vaccination in adolescents is safe. These findings add to, and extend upon, those from a nationwide study in Scotland, reporting no association between vaccination and increased risk of hospital stay for 17 different diagnoses among adolescents [5]. The present study also estimated that 2 doses of mRNA vaccine had about 76% effectiveness against hospitalisation due to COVID-19. Similar estimates were reported in 2 case–control studies of adolescents conducted in the US and Brazil during an omicron-predominant period, where VE against COVID-19 hospitalisation or death was about 80% [6,7]. However, given their design, these studies were unable to determine how common severe COVID-19 is. Therefore, the very low absolute risk of severe disease observed in the present cohort study is an important finding for decision-making concerning the need for COVID-19 vaccination in adolescents. Overall, only 47 individuals, or 7 in 100,000, were hospitalised due to COVID-19 during a period of 5 months, and none of all adolescents hospitalised due to COVID-19 since the start of the pandemic died within 30 days of hospitalisation. Based on this very low risk of severe COVID-19 in the total cohort, the NNV with 2 doses to prevent 1 case during follow-up was more than 8,000. Interestingly, the absolute risk of being hospitalised due to COVID-19 and influenza was similar, despite that the infection pressure from SARS-CoV-2 during follow-up was more than 100 times higher than that of influenza [19].

Given the above, it is important to evaluate whether certain groups of adolescents are at higher risk of severe COVID-19, and if so, whether vaccination is associated with similar protection among these individuals. This study identified 2 different clusters of diagnoses that increased the risk of COVID-19 hospitalisation more than 10-fold. The first included previous infections (bacterial infection, tonsillitis, and pneumonia), and the second included cerebral palsy and other developmental disorders. An encouraging finding was that the VE in these subgroups was similar to that in the total cohort, and consequently, the NNV to prevent 1 case of COVID-19 hospitalisation was about 1,000 individuals. It is therefore of concern that vaccination coverage in these risk groups was somewhat lower compared to in the total cohort. Although we are unable to determine the underlying causes for this observation, factors such as recurrent infections interfering with the administration of the vaccine, and fear of adverse

events, may have contributed. Taken together, these findings suggest that vaccination of adolescents during the omicron era should primarily be targeting those at high risk of severe disease, such as those with previous infections and development disorders.

Concerning the outcome of SARS-CoV-2 infection of any severity, the results indicated that VE from primary vaccination wanes within a few months, similar to observations made in a few other countries [7–9]. However, there is a lack of data exploring whether booster doses reduce the risk of severe COVID-19 as compared to primary vaccination [20]. The results for this comparison in the present study showed that COVID-19 hospitalisations during follow-up were once again extremely rare, implying that administration of booster doses to the general population of adolescents may not be warranted at the current stage of the pandemic. Based on these findings, it is of interest that the US and several countries in Europe are recommending booster doses to adolescents during the omicron era [21,22].

The present study has limitations that should be considered. Because of the observational design, conclusions based on the associations found should be made with caution. For example, despite that VE changed marginally before and after adjustment for covariates, there may be other factors that could have influenced the estimates. However, for the analysis of serious adverse events, we evaluated all diagnoses set during hospital stay both before and after the first dose of vaccine, thereby increasing the chance of detecting selection bias that could interfere with the results. In addition, the sensitivity analysis wherein influenza was used as a negative control outcome supported the lack of important confounding. Moreover, because COVID-19 hospitalisation in this population was very rare, it was not possible to estimate the VE of a first booster dose in this cohort with any form of precision. Finally, another limitation is that even though we excluded all individuals with documented prior SARS-CoV-2 infection, the estimates of VE could be underestimated if some of the individuals in the unvaccinated control group had acquired immunity from a prior SARS-CoV-2 infection that was either asymptomatic or undocumented. Strengths of this study include that all the registers used to obtain the data used in the present study have nationwide coverage with virtually zero loss to follow-up. Finally, the study cohort was based on the total population of Swedish adolescents, including 0.8 million individuals aged 11 to 19 years, which increases the possibility to generalise the findings to other countries with similar population structures.

In summary, monovalent COVID-19 mRNA vaccination was not associated with an increased risk for hospitalisation of any cause in adolescents, suggesting that they are safe to use. While vaccination was associated with a reduced risk of COVID-19 hospitalisation during the omicron era, the absolute risk among the general population of adolescents was extremely low. In contrast, in a large proportion of adolescents hospitalised due to COVID-19, certain risk factors were present, and the effectiveness of vaccination in these individuals was similar to in the total cohort. These results indicate that certain vulnerable subgroups of adolescents, rather than adolescents in general, should be prioritised for vaccination.

## Supporting information

**S1 STROBE Checklist. STROBE Checklist.**
(DOCX)

**S1 Table. Type of SARS-CoV-2 genotypes based on whole genome sequencing in Sweden during the follow-up period of the present study (week 1–22, 2022).** Data publicly available at the Public Health Agency of Sweden (https://www.folkhalsomyndigheten.se/smittskydd-beredskap/utbrott/aktuellautbrott/covid-19/statistik-och-analyser/sars-cov-2-virusvarianter-av-sarskild-betydelse/).
(PDF)

## Author Contributions

**Conceptualization:** Peter Nordström.

**Data curation:** Peter Nordström.

**Formal analysis:** Peter Nordström.

**Investigation:** Peter Nordström, Marcel Ballin.

**Methodology:** Peter Nordström, Anna Nordström.

**Supervision:** Anna Nordström.

**Validation:** Anna Nordström.

**Writing – original draft:** Peter Nordström, Marcel Ballin.

**Writing – review & editing:** Peter Nordström, Marcel Ballin, Anna Nordström.

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
