## [Editor Report · Decision Letter 0]

19 Oct 2022

Dear Dr Nordström, 

Thank you for submitting your manuscript entitled "Safety and effectiveness of COVID-19 mRNA vaccination and risk factors for hospitalisation caused by the omicron variant in 0.8 million adolescents: A nationwide cohort study in Sweden" for consideration by PLOS Medicine.

Your manuscript has now been evaluated by the PLOS Medicine editorial staff as well as by an academic editor with relevant expertise and I am writing to let you know that we would like to send your submission out for external peer review.

Please re-submit your manuscript within two working days, i.e. by Oct 21 2022 11:59PM.

Kind regards,

Philippa Dodd, MBBS MRCP PhD

Senior Editor

PLOS Medicine

---

## [Decision Letter · Decision Letter 1]

7 Dec 2022

Dear Dr. Nordström,

Thank you very much for submitting your manuscript "Safety and effectiveness of COVID-19 mRNA vaccination and risk factors for hospitalisation caused by the omicron variant in 0.8 million adolescents: A nationwide cohort study in Sweden" (PMEDICINE-D-22-03429R1) for consideration at PLOS Medicine. 

[LINK]

In light of these reviews, I am afraid that we will not be able to accept the manuscript for publication in the journal in its current form, but we would like to consider a revised version that addresses the reviewers' and editors' comments. Obviously we cannot make any decision about publication until we have seen the revised manuscript and your response, and we plan to seek re-review by one or more of the reviewers. 

We expect to receive your revised manuscript by Dec 28 2022 11:59PM. Please email us (plosmedicine@plos.org) if you have any questions or concerns.

We look forward to receiving your revised manuscript. 

Sincerely,

Philippa Dodd, MBBS MRCP PhD

PLOS Medicine

plosmedicine.org

GENERAL

Please respond to all editor and reviewer comments detailed below, in full

Please ensure that the study is reported according to the STROBE guideline, and include the completed STROBE checklist as Supporting Information. Please add the following statement, or similar, to the Methods: "This study is reported as per the Strengthening the Reporting of Observational Studies in Epidemiology (STROBE) guideline (S1 Checklist)."

DATA AVAILABILITY STATEMENT

Thank you for including a Data Availability Statement (DAS) which requires revision. For each data source used in your study: 

COMMENTS FROM THE ACADEMIC EDITOR

It would be of value, if the authors explain why they prefer to select matched controls and thereafter adjust for the matching variables - instead of running the full cohort with adjustments. Less risk of introducing bias in the latter alternative. Further, I would like to underline reviewer #3’s request to, if possible, include more potential confounders from available registers.

ABSTRACT

Abstract Background:

Please ensure that the final sentence clearly states the study question

Line 44: Please include a summary of the adverse events referred to 

Abstract methods and findings:

Line 49-50: when reporting statistical information suggest placing a colon after OR instead of a comma, placing CIs within square parentheses and using lower case p, as follows: (OR: 12.0, 95% CI [6.4, 22.6], p<0.001) to improve accessibility to the reader. Please check and amend throughout the manuscript. The use of commas between upper and lower limits adds clarity over the use of hyphens where negative values are reported

Please include any important dependent variables that are adjusted for in the analyses

In the last sentence of the Abstract Methods and Findings section, please describe the main limitation(s) of the study's methodology.

Abstract conclusions:

Line 60: suggest general population of adolescents perhaps? Or something similar

AUTHOR SUMMARY

Thank you for including an author summary

Please change the first sub-heading to “Why was this study done?”

Line 80: starting at “In addition...” suggest this is a separate bullet point

Line 96 onwards: suggest “Clear risk factors for COVID-19 hospitalisation included infections and different developmental disorders, resulting in over a tenfold increased risk of hospitalisation. Vaccine effectiveness among these individuals was similar as in the rest of the cohort.” – I think it might help to elaborate a little on the infections that you mention – what kind? Perhaps would be helpful to broaden the definition in the abstract also

Line 103: as above suggest “general population of adolescents” or something similar. Suggest removing the word “Therefore” and beginning with “Administration…” as a separate bulleted point

Line 105: suggest an individually bulleted point for this statement, which is of significant importance

METHODS and RESULTS

Did your study have a prospective protocol or analysis plan? Please state this (either way) early in the Methods section.

For all observational studies, we ask that authors ensure the following has been clearly indicated in the manuscript text: 

(1) the specific hypotheses you intended to test, 

(2) the analytical methods by which you planned to test them, 

(3) the analyses you actually performed, and 

(4) when reported analyses differ from those that were planned, transparent explanations for differences that affect the reliability of the study's results. If a reported analysis was performed based on an interesting but unanticipated pattern in the data, please be clear that the analysis was data-driven.

Line 194: “….as per when the guidelines for testing in Sweden had changed.” Suggest instead “…in-line with the changes made to testing guidelines in Sweden” or something similar

Please remove “role of the funding source” from the end of the methods section (line 244)

Line 254 onwards: where you report p-values please also report 95% CIs

Line 271: as for the abstract please revise the presentation of statistical information 75% (95% CI [54-86], p<0.001). Please revise throughout to ensure consistency and clarity

Line 306: (VE, 4%, 95% CI, 2-6, 307 P<0.001) perhaps a semicolon following VE? And parentheses and commas for CIs as previously detailed

Line 315: “…with no detectable VE (VE, -6%, 95% CI, -51-119, P=0.88)” is confusing to me, please check and revise accordingly. Please also see reviewer #3 comments.

We agree with the academic editor and the reviewer #3 that adjustment for additional covariates would be beneficial. Please include if possible, if not, please clearly state the reason why.

FIGURES

Line 521 details “Figure legends” but I cannot see any legends in my version of the manuscript or that nay are required, please remove.

Figure 1 – please ensure that all numerical values presented are identical to those in the manuscript text

TABLES

Table 2: Please also present the unadjusted analyses for comparison. As for the adjusted analyses please report 95% CIs and p-values

Table 3 and 4: please also report p-values where you report 95% CIs for adjusted and unadjusted analyses

DISCUSSION

Please remove the sub-heading “conclusions “ such that the discussion reads as single piece of continuous prose ending in the one paragraph conclusion.

Please remove declaration statements from the end of the discussion (line 405, 408, 411) and include only in the submission form when you re-submit your manuscript

REFERENCES

In text reference call outs should be placed within square brackets preceding punctuation as follows: Line 128 – “alleviated their restrictions [1].” As opposed to “alleviated their restrictions.[1]”

Please also remove spaces between citations where more than one study is cited i.e. [1,2,3,4] as opposed to [1, 2, 3, 4]

In the reference list please include up to but no more than 6 author names followed by et al where more than 6 authors contribute to a study

Please use the "Vancouver" style for reference formatting, and see our website for other reference guidelines https://journals.plos.org/plosmedicine/s/submission-guidelines#loc-references

Journal name abbreviations should be those found in the National Center for Biotechnology Information (NCBI) databases. 

Comments from the reviewers:

Reviewer #1: 

This paper describes the vaccine effectiveness and safety of mRNA COVID-19 vaccines among adolescents in Sweden during Omicron predominance, which is an important and timely topic. VE estimates against hospitalization for adolescents during Omicron predominance are needed, and thus the paper adds important evidence. The paper will be improved by additional explanation of methods and, if possible, inclusion of myocarditis as a specific safety outcome. My specific comments are below: 

Specific comments:

1. The authors need to further explain the infection eligibility in the cohort. I don't understand the purpose of this eligibility and matching criteria nor if it was used for an analysis. The authors need to better explain that. This is described as a cohort study, in which case there should not be any selection based on the outcome (SARS-COV-2 infection or hospitalization). 

2. The overall conclusion that adolescents may not need COVID-19 vaccination is based on low absolute risk of hospitalization, but this doesn't account for long-COVID or risk of MIS-C, which should be discussed. Additionally, the VE was 75% for hospitalization, meaning that there was a 75% risk reduction in hospitalization for adolescents after 2 doses of vaccination.

3. Also, the outcome of all-cause hospitalization for safety is important. Can the authors potentially also examine myocarditis/pericarditis for a safety outcome? If they cannot, they should mention myocarditis/pericarditis in the discussion. 

4. Given that bivalent mRNA vaccines are now in use in multiple countries, the authors need to clearly specify in the title and abstract that these data reflect monovalent mRNA vaccines. 

5. Abstract: The sentence regarding vaccine safety should specify that it is all-cause hospitalizations and that vaccinated in that sentence means ≥1 mRNA COVID-19 vaccine dose (which is different from the VE definition).

b. Abstract: Specify what previous infections are risk factors for COVID-19 hospitalizations. Do you mean prior COVID-19 infection as that would be counterintuitive? If this means prior infections, please specify of what and how this is measured. 

c. Abstract conclusion regarding safety states "COVID-19 mRNA vaccination was not associated with an increased risk of any serious adverse event in adolescents" but only data on SAEs associated with hospitalization were presented in the abstract. The conclusion should reflect the data presented. 

6. Summary: regarding this conclusion: "In contrast, individuals with a high risk for severe COVID-19 should be prioritised for vaccination."

a. It might be helpful to state here whether these individuals with a high risk for severe COVID-19 should be prioritized for primary series, additional doses, or both (likely both). 

7. The introduction should include a summary of Sweden's COVID-19 vaccine recommendations for adolescents, so that non-Swedish readers can interpret these data in the context of Sweden's recommendations. 

8. Introduction: Lines 134-135: This sentence implies that the data are inconsistent with myocarditis after COVID-19 vaccine. At this point, the data are fairly clear that there is a rare risk of myocarditis/pericarditis after COVID-19 vaccines (https://www.cdc.gov/vaccines/covid-19/clinical-considerations/myocarditis.html), and the risk is highest in adolescent males (half the population in this study). This sentence should be amended to clarify this point.

9. Introduction: Lines 137-139: Please specify the variant period for this 80% VE against hospitalization from prior studies. 

10. Methods: Lines 164-165: I found this section a bit hard to follow. Were vaccinated adolescents matched to unvaccinated adolescents regardless of infection status? Similarly were adolescents with infection matched to those without regardless of vaccination status? Can you clarify?

11. Lines 244-245: Role of the funding source: This might be better stated as "no external funding was used for this study," as presumably the authors did this as part of their employment (meaning their time was funded by their employers) and the data likely is funded by Sweden's public health system. 

12. Figure 1: 

a. It would be helpful to specifically state how many adolescents were excluded with known prior SARS-CoV-2 infection. 

b. It would be also helpful in the paper to specify what the hospitalization rate was among adolescents with any known prior SARS-CoV-2 infection is, as these adolescents would presumably be at lower risk of hospitalization. Given that in some countries (such as the US), the majority of people in this age group now have had prior COVID-19 infection, this question is of direct public health relevance. 

13. Table 1: It is unclear what the previous ICD-10 diagnoses are from: any previous diagnosis listed in their medical records? Or it is only from a hospitalization? Is this lifetime or over a certain time frame? All that needs to be specified. 

14. Results lines 270-275: Please include a median and IQR for times (days if possible) since vaccination. Given that we know that mRNA VE wanes over time in general, but may be more preserved for more severe outcomes, particularly among younger people (like adolescents), any additional data that can be included to allow us to understand whether these data had any suggestion of waning would be very helpful. 

15. Results lines 276-277: Please include the percent of adolescents with 3 doses who were hospitalized (i.e. 12/41,225 = 0.029%). This percent appears to be higher than for controls (and 2-dose recipients). Is there a reason for this? What was the booster recommendation? How many days had it been since the booster dose? 

16. Results lines 283-285: Among those previously hospitalized for another condition, during what time frame were they hospitalized (over their lifetime versus during a specific timeframe)?

17. Results 285-289: What kind of infection? More specificity on how you defined these ICD codes included in infection needs to be included. What other buckets of previous diagnoses were examined? That is unclear to me. I presume that this is a marker for being higher risk for hospitalization with infections, but do we know why that is? Are these kids with other underlying conditions, such as asthma, technology dependence, immunosuppression, etc? 

18. Results: was there overlap between the group previously diagnosed with infection and those with CP or developmental delay? 

19. Table 2: How were these 30 diagnostic groups defined? Were they from a list of pre-specified potential COVID risk factors or some other list (like complex chronic conditions of childhood) that has been shown to affect risk of hospitalization or death? That needs to be defined. 

20. Table 3: Please include the first date of vaccine eligibility for these adolescents and if possible, median and IQR for days since vaccination.

21. Table 4: Please include the first date of vaccine eligibility for these adolescents (I am guessing it is September 1, 2021) and if possible, median and IQR for days since vaccination.

22. Discussion: Also, what was the infection-induced immunity seroprevalence in Sweden during this time, especially during adolescents? That would helpful add this (or anything known about this) to the discussion to put these results into context.

Reviewer #2: Alex McConnachie, Statistical Review

Nordstrom et al report on a nationwide, retrospective cohort study, looking at the association between COVID-19 mRNA vaccination, and subsequent adverse events and COVID-19 hospitalisations and infections during the omicron wave. This review considers the statistical aspects of the paper.

In short, these are very good. The population, exposures, and outcomes are clearly described. The statistical methods are well described and appropriate. A negative control analysis is also included.

A couple of minor points.

Was the PH assumption checked in the Cox models?

Line 278: I find hyphens don't work in confidence intervals where the values can be negative.

Reviewer #3: Nordström et al. safety of covid-19 vaccination, omicron.

Very important study.

my main objection to this study is that vaccination depends heavily on underlying factors (health seeking behaviour, underlying diseases/comorbidity, perhaps smoking, level of education etc etc), that either a heavily adjusted model or a propensity score model would have been better for this study.

Voluntary/"Elective" vaccinations (e.g. influenza) are often linked to "healthy behaviour" (influenza vaccinations typically protect against death all-year round, even when influenza is not around). if that is the case with covid.19 vaccination, then that would lead to a lower risk of hospitalisation for most things, including covid-19 (the so called vaccine efficiency) in this study.

Abstract, I am particularly happy to see that the authors present absolute risks/numbers needed to treat. I fully agree that needing to vaccinate >9000 teenagers to save 1 hospital admission for covid-19 (with 0 mortality during the Omicron period), argues against general vaccination in this age group.

Research summary; what did the researchers do and fin, please clarify that individuals were followed for 6 months. (study period), to say you followed them until June 2022 says rather little. 

I do not understand the statement that covid-19 vaccination was not associated with risk of "any diagnoses" (??), you mean of any covid-19 diagnoses? or any hospital admission??

the safety analysis is important. However, I am surprised the authors did not specifically look at myocarditis/pericarditis since that is where the worry has been, and they even mention these potential adverse events in the research summary of earlier research.

Ethics approval. While informed consent is often waived in register-based studies that is usually not because of their retrospective study design but because of their register-based nature, that already collected data are used, and that the study is simply not feasible/possible if informed consent was required. So the authors may consider removing the argument why informed consent was waived.

Did really Statistics Sweden identify the exposed? That is usually the job of the National Board of Health and Welfare (or perhaps the Public Health Agency). Matching is however done by Statistics Sweden.

Control selection. there is of course a risk that individuals who chose not to vaccinate themselves did so because they were sure they had had covid-19 (even if not documented), and that would lead to a lower risk of covid-19 among the unexposed (thereby underestimating any protective effect of vaccination) how did the authors tackle this?

I like the dummy analyses with influenza as the outcome.

However, I still feel that any vaccination analysis should be adjusted for additional co-variates. I understand that such covariates may not have been available up until "vaccination date" since there is a lag of reporting some data, but perhaps adjust for covariates up until Dec 31, 2021 in your analyses?

in the results you state that controls were less often diagnosed with a previous SARS-COV2 infection, but was not such infection an exclusion criteria for control eligibility??

Sensitivity analysis, how can the VE 95%CI for influensa be -51 to 119, if the Vaccine efficiency was -6% (should be in the middle of the 95%CI, and if that is percent, how an it exceed 100%)?

Table 1. the codes for infections by no means cover all infections, only a small proportion. 

a) please clarify that ("selected infections"), 

b) add to the table legends what infections were included.

F10 is alcohol dependency, it is "acute alcohol intoxication", typically a young person with binge drinking Friday or Saturday night who ends up in the emergency department. 

If you include Crohn's disease (K50) it is strange that you have not included the more common ulcerative colitis (K51)

Minor Table 1. incorrect spelling of "sinusitis"

[LINK]

---

## [Decision Letter · Decision Letter 2]

23 Jan 2023

Dear Dr. Nordström,

Thank you very much for re-submitting your manuscript "Safety and effectiveness of monovalent COVID-19 mRNA vaccination and risk factors for hospitalisation caused by the omicron variant in 0.8 million adolescents: A nationwide cohort study in Sweden" (PMEDICINE-D-22-03429R2) for review by PLOS Medicine.

I have discussed the paper with my colleagues and the academic editor and it was also seen again by xxx reviewers. I am pleased to say that provided the remaining editorial and production issues are dealt with we are planning to accept the paper for publication in the journal.

[LINK]

We look forward to receiving the revised manuscript by Jan 30 2023 11:59PM.   

Sincerely,

Philippa Dodd, MBBS MRCP PhD

Senior Editor 

PLOS Medicine

plosmedicine.org

Requests from Editors:

GENERAL

Thank you for your detailed and considerate responses to previous editor and reviewer comments. Please respond to further outstanding comments as detailed below

STATISTICAL REPORTING

p values should be reported with a lowercase p. Please amend throughout including tables and figures where relevant

DATA AVAILABILITY STATEMENT

Thank you for updating your statement. Please upload this revised statement into the masncuript submission form when you re-submit your manuscript

AUTHOR SUMMARY

What did the researchers do and find – “There were certain strong risk factors….” Suggest instead, “There were specific risk factors for COVID-19 hospitalization including…”

What do these findings mean – point 2 starting “In contrast…” – suggest making this a separate bulleted point

ABSTRACT

Methods and findings section - “Strong risk factors for …” suggest alternative to the word strong perhaps “Predominant” 

METHODS

Final paragraph – please ensure that (95%) CI has been defined at first use – I couldn’t see that it had but apologies if I have missed it

RESULTS

Please report p values using a lowercase p. Please check and amend throughout

DISCUSSION

Page 2, paragraph 1, final line: “… be targeting risk groups…” suggest “at risk” or “high risk” or something similar

SOCIAL MEDIA

To help us extend the reach of your research, please provide any Twitter handle(s) that would be appropriate to tag, including your own, your coauthors’, your institution, funder, or lab. Please respond to this email with any handles you wish to be included when we tweet this paper.

Comments from Reviewers:

Reviewer #2: Alex McConnachie, Statistical Review

I had very little to criticise in my original review, and the comments I had have been dealt with. I have no further comments.

Reviewer #3: Congratulations on a nice paper.¨

I especially appreciated the use of a negative control outcome: influenza.

[LINK]

---

## [Editor Report · Decision Letter 3]

30 Jan 2023

Dear Dr Nordström, 

On behalf of my colleagues and the Academic Editor, Professor Lars Persson, I am pleased to inform you that we have agreed to publish your manuscript "Safety and effectiveness of monovalent COVID-19 mRNA vaccination and risk factors for hospitalisation caused by the omicron variant in 0.8 million adolescents: A nationwide cohort study in Sweden" (PMEDICINE-D-22-03429R3) in PLOS Medicine.

Prior to publication we require that you make the following final corrections:

1) Please ensure that the following statement regarding your data availability is uploaded into the appropriate section of the manuscript submission form (I could only see the previous version of your statement):

“The data underlying the findings of the present study were used under ethical approval and are publicly unavailable according to regulations under Swedish law. Researchers who are interested in obtaining the data can contact Statistics Sweden via mikrodata@scb.se (https://www.scb.se/en/services/ordering-data-and-statistics/ordering-microdata/), National Board of Health and Welfare via registerservice@socialstyrelsen.se (https://bestalladata.socialstyrelsen.se), and Public Health Agency of Sweden via info@folkhalsomyndigheten.se (https://www.folkhalsomyndigheten.se/the-public-healthagency-of-sweden/)” 

2) Author Summary:

Thank you for your response and for noting my mistake! My apologies, I was meaning to refer to the “What did the researchers do and find?” section. The 2nd bullet point, where the sentence starts “In contrast…” please make this the 3rd of 4 bullet points.

3) Please ensure that your Twitter handles, as detailed to us below, are uploaded into the appropriate section of the manuscript submission form:

“Uppsala University (@UU_University and @uppsalauni) and Umeå University (@UmeaUniversity and @umeauniversitet).”

PRESS

Thank you again for submitting to PLOS Medicine, it has been a pleasure handling your manuscript. We look forward to publishing your paper. 

Best wishes,

Pippa

Philippa Dodd, MBBS MRCP PhD 

PLOS Medicine